# A Statistical Approach to Illustrate the Challenge of Astrobiology for Public Outreach

**DOI:** 10.3390/life7040040

**Published:** 2017-10-26

**Authors:** Frédéric Foucher, Keyron Hickman-Lewis, Frances Westall, André Brack

**Affiliations:** CNRS, Centre de Biophysique Moléculaire, UPR 4301, Rue Charles Sadron, CS80054, 45071 Orléans CEDEX, France; keyron.hickman-lewis@cnrs-orleans.fr (K.H.-L.); frances.westall@cnrs.fr (F.W.); andre.brack@cnrs-orleans.fr (A.B.)

**Keywords:** astrobiology, extraterrestrial life, statistical model, search for life, biosignatures, education and outreach

## Abstract

In this study, we attempt to illustrate the competition that constitutes the main challenge of astrobiology, namely the competition between the probability of extraterrestrial life and its detectability. To illustrate this fact, we propose a simple statistical approach based on our knowledge of the Universe and the Milky Way, the Solar System, and the evolution of life on Earth permitting us to obtain the order of magnitude of the distance between Earth and bodies inhabited by more or less evolved past or present life forms, and the consequences of this probability for the detection of associated biosignatures. We thus show that the probability of the existence of evolved extraterrestrial forms of life increases with distance from the Earth while, at the same time, the number of detectable biosignatures decreases due to technical and physical limitations. This approach allows us to easily explain to the general public why it is very improbable to detect a signal of extraterrestrial intelligence while it is justified to launch space probes dedicated to the search for microbial life in the Solar System.

## 1. Introduction

Although the Earth is the only body known to host life, the number of stars and planets in the Universe encourages us to believe that life appeared elsewhere. The discovery of life arising independently elsewhere would have important consequences for science and, indeed, for humanity. In particular, it would permit better understanding of the origin of life on an Earth whose early geological record has been lost due to crustal recycling and plate tectonics. This quest, the basis of astrobiology [1], is thus one of the most important objectives in the exploration of the Universe. Astrobiology gathers together scientists from various domains, from astronomers to biologists via geologists, through chemists, physicists and philosophers. Due to this very wide and interdisciplinary coverage, and due to the fundamental nature of the discoveries in this domain, astrobiologists are often requested to communicate with the general public through conferences, interviews, or training courses. However, discussing extraterrestrial life with people not directly involved in the domain can rapidly become challenging, largely due to the influence of science fiction in films, books or on the internet upon popular culture. Questions about UFOs, the Roswell Incident and other conspiracy theories are common during these kinds of discourses. It is, therefore, necessary to replace astrobiology in a rigorous scientific framework and to find a simple way to explain the difference between looking for microbial life and looking for extraterrestrial intelligence, without neglecting the possibility of the latter. Here, we present a simple statistical approach permitting us to put into perspective this astrobiological challenge. Based on the Darwinian theory of evolution, it is logical to suppose that the first life was simple microbial life. From this, to reach the stage of intelligent organisms, several consecutive important stages are needed, such as the evolution of photosynthesis, the development of nucleus- and mitochondria-like structures, multicellularity, etc. many stages that require particular environmental conditions. Implicitly, it is thus easy to conceive that simple microbial life would be statistically more common in the Universe than any type of advanced extraterrestrial intelligence. Consequently, considering that inhabited bodies are distributed randomly in space in the Universe, it is implied that highly sophisticated forms of life are statistically further from the Earth than microbial forms of life. On the other hand, if the probability of finding extraterrestrial life and intelligence increases with the distance from the Earth, the number of detectable biosignatures decreases due to technical limitations. This competition between probability and detectability constitutes the main challenge in astrobiology.

In this study, we attempt to illustrate this fact using a statistical approach based on our knowledge of the Milky Way, the Solar System and the evolution of life on Earth. We propose an equation permitting us to obtain the order of magnitude of the probability of life at different stages of evolution and the distance between Earth and the associated bodies in our galaxy. The consequences on the detection of the associated biosignatures are then discussed. This approach allows us to easily explain why it is very improbable that we will be able to detect a signal of extraterrestrial intelligence while, at the same time, it is justified to send space probes dedicated to the search for microbial life in the Solar System. This approach is also an effective way in which to tackle other subjects and notions with general public, among them the vastness of the Universe, the distances in astronomy, the definition of life, and space exploration.

## 2. General Overview and Probabilistic Concept

### 2.1. On the Conditions Required for Life to Appear

There are many definitions of life (see R. Popa, 2004, for a discussion of more than 100 definitions [2]), but the one we use here is the following. Schematically, a living system can be defined as a self-replicating open system capable of (Darwinian) evolution [3,4]. However, when searching for life elsewhere in the Universe, scientists must focus their investigations on detectable forms of life, i.e., life forms that are not too different to us. For chemical and physical reasons (tetravalence, hybridisation, bond energy, etc.), it can be considered that only carbon permits the formation of molecules complex and varied enough to constitute living systems. It is thus generally admitted that most forms of life would be based on organic chemistry and comprised of the elements C, H, N and O, as well as P and S [5].

Before evolution may occur, life must appear. The first hypothesis suggesting the origin of life as a complexification of chemical molecules in the atmosphere of the primitive Earth was proposed by Alexander Oparin in 1924 and then reinforced by the famous experiment of Stanley Miller and Harold Urey in 1953, the latter experiment obtaining amino acids from a gas mixture traversed by an electric arc to simulate lightning [6,7]. Since then, it has been shown that the compositions and concentrations of the gases used were not perfectly relevant to the primitive Earth and that the production of amino acids by this method would have been less efficient. Nevertheless, this particular experiment is the basis of prebiotic chemistry and of exobiology in general. Organic molecules may also form near sea floor hot springs by Fisher-Tropsch reactions [8]. Finally, many molecules have been detected in space [9] and laboratory experiments have shown that a large variety of organic molecules can be formed in ice [10,11]. They are also found in comets, as recently shown by the Rosetta mission [12], and in meteorites [13]. The Murchison meteorite contains several thousands of different organic molecules, in particular 70 amino acids of which 8 are proteinogenic [14]. That so many pivotal organic molecules exist and survive in space has given credence to the Panspermia hypothesis, and further fuels postulation that life may be able to undergo transport through space on scales larger than that of a single planet [15].

Chemists considered that the first ‘living systems’ were formed on Earth by a suite of chemical reactions involving these different molecules. For these reactions to occur, liquid water played a key role, in particular due to its exceptional properties, for instance of allowing for the stability of key chemical bonds to create and maintain chiral macrostructures while favouring the breakup of other bonds to permit chemical interchange [5,16,17]. Indeed, all known living organisms on Earth are assemblages of carbon molecules containing liquid water. Prebiotic chemistry experimentation and modelling is continuously increasing our understanding of the reactions leading to the formation of the building blocks of life: nucleic acids and bases, amino acids, and lipids [18]. In particular, present studies in this domain now focus their investigations on the reaction of organic molecules in liquid water in contact with mineral surfaces sensu lato, in what is commonly called the primordial soup (see e.g., [17,19]). The energy source needed for the reactions to occur could have been light from the Sun, a geological heat source (thermal gradient associated with hydrothermal systems, and stemming from the heat flux of the cooling Earth), radioactivity (now-extinct radionuclides responsible for much heat flux on the early Earth), or another, unknown, cause. Considering all of the above, one can conclude that a rocky body hosting liquid water that can be sown with organic molecules by meteoritic input and/or Fischer-Tropch type reactions, and where these products can be reprocessed in chemical reactors in a presence of a source of energy, is needed for life to appear and to develop. These conditions may appear to be relatively strict but are in reality common in the Universe. Moreover, they apply to life forms with which we are familiar (i.e., Terrestrial life) and thus for life as we seek it. Note, however, that astrobiologists do not exclude the possibility of unexpected types of life [20].

### 2.2. On the Probability of the Occurrence of Extraterrestrial Life

The statistical approach is the best way to estimate the probability of extraterrestrial life. Thus, Frank Drake tried to estimate the number of civilisations in the Milky Way capable of emitting a detectable sign into space, and willing to communicate, using his now-famous equation [21]. The Drake equation is the product of several probabilistic terms, most of them difficult to evaluate [22]. Depending on the specificities of the approach, the number of extant intelligent civilisations capable of communication in the Milky Way may be lower than one, meaning that the existence of intelligent life forms is very unlikely and that the number of extraterrestrial civilisations is lower than the number of galaxies in the Universe; alternatively, it may be of the order of several tens of thousands. Many studies attempting to solve this equation have been published, some with specific focus on particular terms of the equation or the use of new approaches based on different mathematical models (e.g., [23,24,25,26,27,28,29]). The Ozma project (1960) and the Frank Drake study (1961) are at the origin of the Search for Extra-Terrestrial Intelligence (SETI) programme looking for intelligent signals using, in particular, the Arecibo telescope to collect data for the seti@home project (http://setiathome.berkeley.edu/).

Nevertheless, in astrobiology, the search for extraterrestrial life is not limited to intelligent civilisations. On the contrary, research in this domain mostly focuses on microbial life, particularly with respect to the Solar System. Thus, S. Seager undertook a broader approach to investigate the detectability of life on exoplanets located in the habitable zone of their star [30,31]. In her study, the author attempted to calculate the number of planets upon which life emerged and developed sufficiently to change the atmosphere by the emission of biogases, yet still near enough to be detected from the Earth. More recently, D. Waltham estimated the probability of reaching more or less evolved life on habitable Earth-like exoplanets [32]. Note, however, that it is theoretically possible for microbial life to appear even if the host body is not globally habitable [33]. At the microbial scale, recent exploration of the Solar System has shown that environments compatible with the existence of life may be relatively common in the Universe and it is widely believed that life may have appeared on other bodies in the Solar System, for example below the icy crust of the moons of Saturn and Jupiter, or on Mars [34,35,36]. Detecting potential remains or traces of life will be one of the aims of in situ investigations of the Martian surface made by the future ESA/Roscosmos ExoMars 2020 rover, and by the NASA Mars 2020 mission.

If life appeared on a solid body other than the Earth in the Solar System, this should mean that life is very common in the Universe, statistically leading to a strong increase in the number of potential civilisations estimated by the Drake equation. Concomitantly, the absence of traces of life on another ‘habitable’ body in the Solar System does not mean that life is unique to the Earth, but it does mean that it is potentially rarer in the Universe. Here, we extend the statistical approach used by F. Drake, S. Seager and D. Waltham [21,30,31,32] to the concept of life in general in order to include the search for past or present microbial life in the Solar System.

### 2.3. The More Complex, the Less Probable

Defining the complexity of an organism is not obvious. Indeed complex, self-organising structures and organisms do not require complex mechanisms of formation [37], nor should simple structures and organisms be the expected results of simpler processes. Darwinian evolution is a continuous process that leads equally to the diversity of life, as well as to its complexity. However, for general public outreach, it is useful to illustrate the complexity of life by using particular forms of life that are representative of chronological stages of evolution. We choose : (1) microbial life forms similar to unicellular prokaryotes, (2) macroscopic multicellular organisms as defined by J. Kasting [38], i.e., similar to those of the Ediacaran biota, and (3) intelligent forms of life (civilisations). Throughout this study, after each calculated term, the stage of evolution to which that term applies will be indicated in brackets by (Mic.), (Mac.) and (Civ.) respectively. The global approach to estimate the probability of extraterrestrial life consists of solving an equation composed of several terms that are: (i) a number of particular stars or planets of interest in our galaxy or in the Universe, (ii) a proportion of these stars or planets meeting particular criteria, or (iii) a probability that a particular event occurred, such as the appearance of life. With increasing astronomical knowledge, it is now possible to make a reasonable estimation of the value of the numerical and proportional terms regarding other stars and exoplanets. The Solar System can also be used as a model to estimate some proportional terms related to bodies too small to be observed in other stellar systems.

## 3. Equations for Life in the Universe

### 3.1. Probability Equation for Life in the Universe

We define the proportion of bodies presently hosting one of the three stages of life $P_{Life}$ as a product of different terms:(1)$$P_{Life}=p_{S}\times p_{B}\times p_{C}\times p_{L}\times p_{R}\times p_{P}$$
with:$p_{S}$: the proportion of stellar systems having a star compatible with the occurrence of the considered stage of life.$p_{B}$: the proportion of these stellar systems having at least one rocky body located at a distance from the star compatible with the considered stage of life, i.e., within its habitable zone.$p_{C}$: the proportion of these bodies compatible with the emergence of life.$p_{L}$: the probability that life appeared on these bodies.$p_{R}$: the probability that life reached the considered evolutionary stage on these bodies.$p_{P}$: the probability that life at this evolutionary stage is still active on these bodies. This is equivalent to the probability of co-existence occurrence of this stage of life on several bodies simultaneously.

The resolution of Equation (Equation 1) is fully dependent on the considered evolutionary stage of life (microbial, complex, or intelligent); the values attributed to the different terms are not the same for intelligent forms of life and for microbial life.

### 3.2. Converting Probabilities into Distances

There are about 200 billion stars in the Milky Way. The number $N_{Life}$ of bodies in our galaxy hosting the considered stage of life is given by:(2)$$N_{Life}=P_{Life}\times 200\times 10^{9}$$

Using the value of $N_{Life}$ given by Equation (Equation 2), it is possible to deduce the average distance $D_{Life}$ between two of the considered bodies. The Milky Way is a flattened disk approximately 100,000 light years (ly) in diameter and 6000 ly in thickness. Its volume is thus equal to:(3)$$V_{M.W.}=4.7\times 10^{13}\;{\mathrm{ly}}^{3}$$

The average distance $D_{Life}$ between two potentially inhabited planets is therefore:(4)$$D_{Life}=\sqrt[{3}]{\frac{V_{M.W.}}{N_{Life}}}$$

It is important to note that this distance may vary due to the inhomogeneity of the galaxy. Indeed, the density of stars is, for instance, higher in the arms of the galaxy and globally increases with the decreasing radial distance from the centre of the galaxy. The distance $D_{Life}$ would thus be lower in these parts of the galaxy and most of the habitable planets are expected to be located in the inner Galaxy [39]. On the other hand, our Solar System is located at an average distance from the centre of the Milky Way, in the arm of Orion. The distance given by Equation (Equation 4) can thus be considered as representative of the average distance between the Earth and other considered inhabited planets.

In the following, the values of $P_{Life}$, $N_{Life}$ and $D_{Life}$ are estimated for the three considered stages of life.

## 4. Solving the Equations

### 4.1. Astronomical Terms

The three first terms in Equation (Equation 1) are representative of astronomical parameters. They can be estimated with a relatively good accuracy based on our knowledge of the Universe and data coming from various telescopes.

$p_{s}$ is the proportion of stellar systems having a star compatible with the occurrence of the considered stage of life. The essential requirements of prokaryote-like unicellular life are relatively simple; this kind of life form does not require as much energy or nutrients as macroscopic multicellular organisms in order to flourish. While being broadly globally distributed on Earth, certain strains of microbes can also develop patchily, for instance in ecological niches at the sub-millimetre scale [33]. Therefore, it is expected that prokaryote-like life could develop at the surface or subsurface of planets or satellites. Most stars are thought to have at least one planet [40], and many planets are thought to have satellite(s) [41]. Even if the presence of extraterrestrial life is not probable around massive blue giants due to their very short lifetime, exoplanets have been found around them [42] and these systems can thus potentially host habitable niches without requiring that life had the time to appear. Thus, we can consider that more or less all stellar systems have bodies compatible with prokaryote-like unicellular life and $p_{S}\left(Mic.\right)$ can be approximated to 100%.

In contrast to this, multicellular macroscopic organisms only appeared on Earth after several billion years of evolution, and only after the rise of atmospheric oxygen in the Great Oxygenation Event [43,44], implying that photosynthetic organisms played a key role in the rise of multicellularity. Using a restrictive approach, it is thus possible to consider that multicellular macroscopic organisms may only develop under the influence of a star small enough to be stable for several billions of years, but large enough to produce the energy required, i.e., ‘Sun-like stars’ [45]. This is ever truer for extraterrestrial civilisations. In the Milky Way, with the proportion of ‘Sun-like stars’ being around 10%, it is possible to set $p_{S}\left(Mac.\right)=p_{S}\left(Civ.\right)=10\%$.

$p_{B}$ is the proportion of previous stellar systems having at least a rocky body located at a distance from the star compatible with the considered stage of life.

To be habitable at the global scale, a planet must be located in the habitable zone of its star, i.e., where the liquid water can be stable at the surface [46,47]. Petigura et al., howed that about 22% of Sun-like stars would have a planet located in the habitable zone [45]. The term $p_{B}$ for macroscopic life and intelligent civilisations can thus be set $p_{B}\left(Mac.\right)=p_{B}\left(Civ.\right)=22\%$. In contrast, for unicellular prokaryote-like organisms that can develop in ecological niches in subsurface of any rocky body (planets or icy moons) where there is liquid water, the concept of a habitable zone around a star being restricted to bodies with stable liquid water at the surface is no longer valid. The habitable zone becomes so vastly extended that $p_{B}\left(Mic.\right)$ can be set at $p_{B}\left(Mic.\right)=100\%$.

$p_{C}$ is the proportion of the previous bodies compatible with the emergence of life. In the Solar System, the smallest body upon which life may have appeared is Enceladus, a moon of Saturn [36]. There are 8 planets and 19 moons larger than or equal in size to Enceladus in the Solar System and six of them (the Earth, Mars, Enceladus, Europa, Ganymede, and Titan) have, or had, environmental conditions compatible with the emergence of life, i.e., 22% of them. Using the Solar System as an example, we can set $p_{C}\left(Mic.\right)=22\%$.

The planets located in the habitable zone of their star are not necessarily compatible with complex life. In particular, atmospheric evolution would play a key role in the habitability of a planet through time. For instance, a planet could lose its atmosphere with the consequence that liquid water would no longer be stable at its surface (similar to Mars), or a planet may have environmental conditions that are incompatible with the emergence of life (Venus, for instance). The observation of exoplanets located in the habitable zone of their star is still relatively limited and the proportion of planets located in the habitable zone of Sun-like stars that are truly habitable is difficult to determine. However, of the confirmed exoplanets located in the habitable zone of their star, only ‘mesoplanets’, i.e., planets where the temperature at the surface is potentially between 0 and 50 °C, are compatible with complex life. Our knowledge about surface properties of exoplanets is very limited, however, following https://fr.wikipedia.org/wiki/Liste_d’exoplanètes_potentiellement_habitables, we can roughly estimate this proportion to be 75%. The terms $p_{C}\left(Mac.\right)$ and $p_{c}\left(Civ.\right)$ can thus be set to 75%. This term appears to be higher for macroscopic multicellular life and extraterrestrial civilisations than for prokaryote-like organisms but this can be explained by the fact that it applies to planets located in the habitable zone in the case of complex and intelligent life, while for it applies to any rocky bodies in the case of prokaryotic life.

### 4.2. Empirical Terms

The main problem of astrobiology is the fact that the only known life is life on Earth. This unique example engenders the biggest challenge of our approach, which is to estimate the order of magnitude of the probabilistic terms related to life itself corresponding to the last three terms in Equation (Equation 1). What is the probability for life to appear on a habitable planet? What is the probability that life evolves into macroscopic multicellular organisms? And into intelligent organisms? C. Maccone described Darwinian Evolution as an exponential growth of evolution with time, leading to an increase in the probability of extraterrestrial civilisations with time [26,27,28,29]. In the present model, less mathematically developed than that of C. Maccone, we also based our approach on the fact that, with increasing time, there is more possibility for life to evolve. Using life on Earth as a reference, we thus considered that the probability to reach each of the three considered stages of evolution is inversely proportional to the ratio of the time they spent to appear on Earth over the time in which they could have appeared. This approach, which consists in considering that the time span between when conditions are required for an event to occur and the moment when it effectively occurs provides a metric of the probability of this event, has been used before to evaluate the probability of abiogenesis based on the history of life on Earth and has been discussed by Spiegel and Turner (2012) [48]. Similarly, we also considered that the probability of coexistence of a given stage of evolution corresponds to the ratio of the time it would have been present on the Earth over the total existing time of the Earth. This approach is illustrated in Figure 1. In order to estimate probability using this method, several examples would normally be required. Indeed, our approach amounts to saying that life on Earth is the paragon of life in the Universe, which is extremely hypothetical, and even optimistic. Furthermore, if the chemical processes leading to abiogenesis can be considered as relatively ‘simple’, evolution of life is very sensitive to the environment and the different stages of evolution considered in this study are the consequences of very specific properties and events that occurred during the history of the Earth such as the formation of the Moon, plate tectonics, major extinction events, etc. Our approach thus demands more speculation as the stage of evolution increases. In any case, it is an easy way to obtain orders of magnitudes that can be discussed later.

$p_{L}$ is the probability that life appeared on the bodies where the considered stage of life could appear. For habitable planets, if we look at the Earth, we could be tempted to set the probability of appearance of life to 100%. However, as explained by C.S. Cockell, a habitable planet does not necessarily mean an inhabited planet [49,50]. The proportion $p_{L}$ of habitable planets where life indeed appeared is thus not necessarily equal to 100%. If we look at the Solar System, there is presently only one planet in the habitable zone and it was inhabited about 300 million years after its formation 4.543 Ga ago. It is also well established that there was surface liquid water 4.3 Ga ago [51]. Moreover, it was showed that the Earth will remain in the habitable zone of the Sun for a further 1 billion years [52]. Ergo, life appeared after 300 million years during the total 5.3 billion years of Earth habitability, i.e., approximately at 6% of that duration. Using our approach, since the probability of occurrence of that event is considered to be inversely proportional to the time it took to occur, we consider that the probability of appearance of life of any type on a habitable planet is $p_{L}\left(Mac.\right)=p_{L}\left(Civ.\right)=94\%$. Due to our approach itself, the rapid appearance of life on Earth leads to this very high value of $p_{L}$ for habitable planets. However, and as stated previously by [48], the rapid appearance of life on Earth does not necessarily mean that the same process of abiogenesis occurred with equal rapidity elsewhere.

For prokaryote-like life potentially inhabiting ecological niches, life on Earth cannot be used as a model since, even it is chemically plausible, we do not know if life actually appeared, for example in the deep sea of Europa. It is thus preferable to advocate a conservative view saying that out of the six bodies of the Solar System where life could have appeared, it only appeared on the Earth; we thus set $p_{L}\left(Mic.\right)=$ 17%.

$p_{R}$ is the probability that life reached the considered stage on the considered bodies. For prokaryote-like life it is thus 100% (if life emerged, it emerged as prokaryotic life, as per our model) and $p_{R}\left(Mic.\right)=100\%$. On Earth, the first occurrence of proposed macroscopic multicellular life has been described from 2.1 Ga old black shales in Gabon [53]. This is debated but we can consider that multicellular life began to undeniably proliferate during the Ediacaran, 600 Ma ago, i.e., after 3.8 billion years out of the 5.3 billion years of Earth’s habitability, i.e., approximately after 72% of that duration. Following the same reasoning as above, we can thus set $p_{R}\left(Mac.\right)=28\%$. Finally, Homo sapiens, which developed the first intelligent civilisation on Earth, appeared 200 000 years ago, i.e., after 4.3 billion years out of the 5.3 billion years of Earth’s habitability, i.e., approximately at 81% of that duration. Following the same reasoning as above, we can thus set $p_{R}\left(Civ.\right)=19\%$.

$p_{P}$ is the probability that the considered life is still presently active on the bodies in question. As stated previously, it is equivalent to the probability of life having reached a given stage of evolution on two bodies simultaneously. Prokaryotes are known to be very resistant and to have a very great capacity for adaptation to environmental stresses at both the local and regional scale. Only a global change affecting the whole considered body, both at the surface and in the subsurface, would lead to the disappearance of prokaryote-like unicellular life. We can thus consider that microbial life could survive until the end of the life of the stellar system in which they developed. On Earth they would then have been present during about 9 billion years over the estimated 10 billion years of life time of the Solar System, before the Sun becomes a Red Giant. We can thus set $p_{P}\left(Mic.\right)=90\%$.

Similarly, since it appeared on Earth, macroscopic multicellular life never completely disappears despite catastrophic events, but rather changes its distribution, mode and faunal hierarchy [54,55]. If we consider that macroscopic life will remain on the Earth until it remains habitable at its surface, for another further 1 billion years, the Earth’s surface would have been inhabited by macroscopic multicellular life during about 1.6 billion years out of the 10 billion year lifetime of the Solar System. We can thus set $p_{P}\left(Mac.\right)=16\%$.

The duration of a civilisation is difficult to evaluate. If we decide to be very conservative (or even pessimistic) and state that humanity might disappear today, from the emergence of Homo sapiens, it would have existed during 2 × $10^{-3}$% of the life time of the Solar System. We can thus set $p_{P}\left(Civ.\right)=0.002\%$.

### 4.3. Results

#### 4.3.1. Prokaryote-Like Unicellular Life

From Equation (Equation 1), the proportion of stellar system hosting prokaryote-like unicellular life is given by:(5)$$P_{Life}\left(Mic.\right)=1\times 1\times 0.22\times 0.17\times 1\times 0.9=3.37\%$$

The corresponding number of bodies inhabited by prokaryote-like unicellular life in the Milky Way is estimated using Equations (Equation 2) and (Equation 5):(6)$$N_{Life}\left(Mic.\right)=6.73\times 10^{9}$$

The associated average distance between two stellar systems inhabited by prokaryote-like unicellular life is given by Equations (Equation 4) and (Equation 6): (7)$$D_{Life}\left(Mic.\right)=19\;\mathrm{ly}$$

#### 4.3.2. Macroscopic Multicellular Life

From Equation (Equation 1), the proportion of stellar systems hosting macroscopic multicellular life is given by:(8)$$P_{Life}\left(Mac.\right)=0.1\times 0.22\times 0.75\times 0.94\times 0.28\times 0.16=0.07\%$$

The corresponding number of planets inhabited by macroscopic multicellular life in the Milky Way is estimated from Equations (Equation 2) to (Equation 8):(9)$$N_{Life}\left(Mac.\right)=1.39\times 10^{8}$$

Equations (Equation 4) and (Equation 9) permit us to deduce the average distance between two planets inhabited by macroscopic multicellular life:(10)$$D_{Life}\left(Mac.\right)=70\;\mathrm{ly}$$

#### 4.3.3. Intelligent Civilisations

Finally, the proportion of stellar systems hosting intelligent civilisations is given by:(11)$$P_{Life}\left(Civ.\right)=0.1\times 0.22\times 0.75\times 0.94\times 0.19\times 0.00002=6\times 10^{-6}\%$$

Thus, after Equations (Equation 2) and (Equation 11), the estimated number of present civilisations in our galaxy is:(12)$$N_{Life}\left(Civ.\right)=11,788$$

This number is in the upper estimate of the value found by solving the Drake equation. However, in our model we considered intelligent civilisation to be approximated as ‘humanity since the appearance of *Homo sapiens*’, whereas the Drake equation considered ‘humanity since its ability to communicate by radio signal’. We made this calculation only for the Milky Way but there are ≈350 billion galaxies in the Universe, each containing approximately the same number of stars. The number of intelligent civilisations in the Universe could thus reach several thousand billion.

Using Equations (Equation 4) and (Equation 12), it is possible to deduce the average distance between two intelligent civilisations in the Milky Way, that is to say the average distance from the Earth:(13)$$D_{Life}\left(Civ.\right)=1586\;\mathrm{ly}$$

#### 4.3.4. Considering Past Life

Multicellular organisms may disappear but their traces may survive until the end of the life of the stellar system in which they developed. On Earth these traces would therefore have been present and available for detection over about 5.6 billion years out of the estimated 10 billion year lifetime of the Solar System. We can thus set $p_{P}{\left(Mac.\right)}_{Past}=56\%$. The proportion of stellar systems hosting past macroscopic multicellular life is then given by:(14)$$P_{Life}{\left(Mac.\right)}_{Past}=0.1\times 0.22\times 0.75\times 0.94\times 0.28\times 0.56=0.24\%$$

The corresponding number of planets in the Milky Way from Equations (Equation 2) to (Equation 14) is:(15)$$N_{Life}{\left(Mac.\right)}_{Past}=4.86\times 10^{8}$$

Equations (Equation 4) and (Equation 15) permit us to deduce the average distance between two planets that are or have been inhabited by macroscopic multicellular life: (16)$$D_{Life}{\left(Mac.\right)}_{Past}=46\;\mathrm{ly}$$

Similarly, civilisations could have also disappeared. On Earth these traces would then have been present during about 5 billion years over the estimated 10 billion years of life time of the Solar System. The probability that past traces of civilisation exist around a star is then given by setting $p_{P}{\left(Civ.\right)}_{past}=0.5$ in Equation (Equation 1). This leads to:(17)$$P_{Life}{\left(Civ.\right)}_{Past}=0.1\times 0.22\times 0.75\times 0.94\times 0.19\times 0.5=0.15\%$$

Equations (Equation 2) and (Equation 17) lead to the estimated number of past civilisations in our galaxy:(18)$$N_{Life}{\left(Civ.\right)}_{Past}=2.95\times 10^{8}$$

Traces of past civilisations could thus be present relatively ‘close’ to the Solar System since the average distance between two past civilisations is, using Equations (Equation 4) and (Equation 18):(19)$$D_{Life}{\left(Civ.\right)}_{Past}=54\;\mathrm{ly}$$

This value is very low meaning that, potentially, traces of more or less developed past civilisations could be common in the Milky Way.

### 4.4. Summary

The results of Equations (Equation 5)–(Equation 19) are compiled in Table 1. Of course, these values must be considered with caution and not as ‘true’ values. The key point here is to compare the order of magnitude of the different probabilities. The probability that there is prokaryote-like life around a star is about 48 times higher than the probability that there is active macroscopic multicellular life and 571,000 times higher than the probability that there is an active civilisation. On the contrary, the probability of finding relics of a past civilisation on an exoplanet is only 23 times lower than the probability to find microbial life. This is intuitively sensible from what we know of the terrestrial fossil records of bacteria versus the fossil records of animals, versus the archaeological records of humans.

Figure 2 describes the schematic evolution of life on a habitable planet from its origin to the disappearance of its stellar system. The green arrows emphasise that, for life of a higher evolutionary stage, the timescale for which it exists decreases dramatically. On the other hand, the yellow arrows show that the duration of existence of the associated traces of past life could, in all cases, potentially exist for a long time.

## 5. On the Detection of Extraterrestrial Life

The search for life in the Universe is driven by the detection of biosignatures. Biosignatures are either more or less obvious depending on the life form that produces these signatures. For example, life is axiomatic on Earth because of the full range that exists from simple microbial to complex intelligent life, and their plethora of activities at the global scale. On the contrary, given the inclement habitable conditions, potential life on Mars would be prokaryote-like and would not have evolved sufficiently to have left a global signature, for example atmospheric oxygen produced by photosynthetisers [33,56]. Finally, biosignatures associated with past traces of life can be altered or even destroyed with time, in particular those associated with microbial life, which have been vastly reduced following the rise of the benthic biome at the Cambrian Explosion of animal life. The techniques involved in the detection of biosignatures are thus all the more ‘sophisticated’ since they are concerned with detecting primitive life forms. The use of sophisticated, lab-based equipment is increasingly necessary to demonstrate the biogenicity of fossilised traces of microbial life. Importantly, with increasing distance from the Earth the number of techniques that can be used for biosignature detection rapidly decreases and, for those that are compatible with planetary exploration, their resolution and capabilities are generally decreased. These technical limitations have important consequences on the detection of extraterrestrial life.

### 5.1. Search for Extraterrestrial Prokaryote-Like Life

While the probability that prokaryote-like life exists elsewhere in the Milky Way is very high, microbes, particularly the most primitive chemosynthetic forms, are associated with subtle biosignatures. The search for extraterrestrial microbial life outside the Solar System is limited to life forms that have evolved enough to change the atmosphere of the planet in order to be detected by spectroscopy. The simultaneous presence of gases forming an unstable mixture (H_2_O, O_2_, O_3_, CH_4_, N_2_O, etc.) may be a good indicator of life [57]. In particular, large amounts of O_2_ and O_3_ can be considered as a signature of oxygenic photosynthetic activity [58] noting, at the same time, that small amounts of these gases can be produced abiotically [59]. The probability of such a kind of life is therefore that of an intermediate stage between microbial unicellular and macroscopic multicellular life (Figure 2). The appearance of oxygenic photosynthetic life capable of changing the whole atmosphere of a planet sufficiently to be detected from the Earth requires that the surface of the planet is habitable, i.e., a planet located in the traditional habitable zone. It is then necessary to recalculate the different terms in Equation (Equation 1). The first terms $p_{S}$, $p_{B}$, $p_{C}$ and $p_{L}$ in Equation (Equation 1) are the same for surface photosynthetic microbial life and macroscopic multicellular life, i.e., 10%, 22%, 75% and 94% respectively. Although it probably appeared much earlier [60,61], oxygenic photosynthetic life of Earth induced detectable changes in the atmosphere about 2.4 Ga ago with the Great Oxygenation Event, i.e., after 1.9 billion years out of the 5.3 billion years of Earth’s habitability, i.e., approximately at 36% of that duration. Following the same reasoning as before, we can thus set $p_{R}{\left(Mic.\right)}_{Photo}=64\%$.

Similarly, if we consider that surface photosynthetic microbial life will remain on the Earth until it is no longer habitable at its surface, we can estimate $p_{P}{\left(Mac.\right)}_{Photo}=34\%$. The proportion of stellar systems potentially hosting detectable photosynthetic life is then:(20)$$P_{Life}{\left(Mic.\right)}_{Photo}=0.1\times 0.22\times 0.75\times 0.94\times 0.64\times 0.34=0.34\%$$

The corresponding number of planets is:(21)$$N_{Life}{\left(Mic.\right)}_{Photo}=6.75\times 10^{8}$$

Using this value, we can deduce the corresponding average distance $D_{Life}{\left(Mic.\right)}_{Photo}$:(22)$$D_{Life}{\left(Mic.\right)}_{Photo}=41\;\mathrm{ly}$$

At this distance, it is possible to detect putatively biologically produced gases in the atmosphere of an exoplanet [62].

If we look for microbes that are only detectable by in situ exploration, our investigations are limited to the Solar System. Of particular interest are the icy moons of Jupiter and Saturn, and the Martian subsurface. The proportion of stellar systems statistically inhabited by prokaryote-like life, given by Equation (Equation 5), is equal to 3.37%. This value can also be seen as the probability for a stellar system to host life. Since life appeared on Earth, it is thus necessary to consider the probability of a second origin of life in the same stellar system, corresponding to the square of Equation (Equation 5). The probability that life appeared on another body in the Solar System is thus only $P_{Life}{\left(Mic.\right)}_{SS}=0.11\%$.

This probability is relatively low. Moreover, if it exists, active life is expected to be present below the icy crust of the moons of Jupiter and Saturn, or several hundred meters deep in the subsurface of Mars [63]. The exploration of the interior of these bodies to search for microbial life is very complicated and will not be possible in the near future. Presently, the investigations focus on the search for past traces of microbial life at the Martian surface.

The existence of prokaryote-like microfossils on the surface of Mars necessarily presupposes the presence of microbial life at its surface in the past. This restriction modifies the value of $p_{C}$ (the proportion of the considered bodies compatible with the emergence of life at their surface). The conditions necessary for the emergence of life on the Martian surface were present during the Noachian [56,63]. The first terms $p_{S}$, $p_{B}$, $p_{C}$ and $p_{L}$ in Equation (Equation 1) are the same for surface microbial life and macroscopic multicellular life, i.e., 10%, 22%, 75% and 94% respectively and $p_{R}{\left(Mic.\right)}_{Surf}=100\%$, as previously determined for microbial life. On Earth, traces of prokaryote-like life would have been present during about 9.5 billion years over the estimated 10 billion year lifetime of the Solar System. We can thus set $p_{P}{\left(Mic.\right)}_{Surf}=95\%$. As for potential active microbial life in the subsurface of icy moons, we therefore need to consider the probability for a second appearance of life in the Solar System. Consequently, the probability that there are microfossils at the surface of Mars is given by:(23)$$P_{Life}{\left(Mic\right)}_{Mars-Past-Surf.}=0.1\times 0.22\times 0.75\times 0.94\times 1\times 0.95\times 0.337=0.05\%$$

This probability is not very high but can justify the launch of instruments dedicated to the search for traces of past prokaryote-like life on Mars surface.

### 5.2. Search for Macroscopic Multicellular Life

Planets potentially inhabited by macroscopic multicellular life could be very common in the Milky Way. However, their distance from the Earth could be large: 70 ly as estimated by Equation (Equation 10). At this distance, direct observation is totally impossible and the only way to detect this kind of life is to search for modifications of the atmosphere of the planet using IR spectroscopy. The presence of forests (i.e., photosynthesising vegetation on exposed land masses) could also be confirmed using IR spectroscopy by the vegetation’s Red Edge due to chlorophyll [64], even if this signature is not unambiguous [65] and if this kind of surface feature would be even more challenging to detect than atmospheric signatures.

On Earth, we know that vegetation has covered much of the exposed landmass for approximately 400 Ma, since the Devonian, i.e., after 4.1 billion years over the 5.3 billion years of Earth habitability, i.e., after approximately 77% of that duration. Following the same reasoning as before, we can thus set $p_{R}{\left(Mac.\right)}_{Veg}=23\%$. Similarly, $p_{P}{\left(Mac.\right)}_{Veg}=14\%$. The proportion of stellar systems hosting vegetation is then:(24)$$P_{Life}{\left(Mac.\right)}_{Veg}=0.1\times 0.22\times 0.75\times 0.94\times 0.23\times 0.14=0.05\%$$

The corresponding number of planets inhabited by vegetation in the Milky Way is, from Equations (Equation 2) to (Equation 24):(25)$$N_{Life}{\left(Mac.\right)}_{Veg}=10\times 10^{7}$$

Equations (Equation 4) and (Equation 25) permit us to estimate the average distance between two planets whose exposed crust is colonised by vegetation:(26)$$D_{Life}{\left(Mac.\right)}_{Photo}=78\;\mathrm{ly}$$

Direct imaging of exoplanets will be needed to detect surface signatures like vegetation’s Red Edge. Limited direct imaging of the surfaces of nearby rocky planets may be possible with the James Webb Space Telescope. However, even if the distance of 78 ly given by Equation (Equation 26) is reasonable for detection, it is probable that more powerful telescopes would be needed to really open the door to such observations.

### 5.3. Search for Extraterrestrial Civilisations

Equation (Equation 13) results in an average distance between two extant extraterrestrial civilisations of 1586 ly. At this distance, the only detectable biosignature would be a radio signal, as previously envisaged by Cocconi and Morrison in 1959 [66] and independently by F. Drake in 1965 [21], or an optical signal, as proposed by Schwartz and Towne in 1961 [67]. In order to be detected, intelligent life must thus have reached a higher echelon of evolution within ‘intelligent life’, corresponding to space communicating life. SETI research is now looking for signals in a large part of the electromagnetic spectrum, from gamma ray to radiowave. Moreover, the detection of exoplanets and the possibility to analyse their atmospheres permit envisaging the detection of new indirect ‘technomarkers’ associated with extraterrestrial civilisations not necessarily capable of and/or willing to communicating in space, such as pollutant molecules in atmospheres [68], or in the aberration of light curves due to megastructures [69]. Finally, recent works also envisaged the detection of life as we do not know it, thus increasing of the potential biosignatures associated with extraterrestrial life [70].

The proportion of stellar systems inhabited by detectable civilisations is thus lower than the number previously calculated, since the terms $p_{R}$ and $p_{P}$ were determined by considering a definition of intelligent life from the appearance of *Homo sapiens*, and not from the appearance of radio communication. On Earth, intelligent life has been able to communicate at distance for only 100 years. With respect to the age of the Earth, the difference in time between the appearance of *Homo sapiens* and the 19th century is too small to change the term $p_{R}$ significantly. On the other hand, with respect to the timescale of human evolution up to the modern day, the time taken for an intelligent civilisation to be able to communicate in space strongly decreases the term $p_{p}$ giving the probability of simultaneity. Taking a pessimistic standpoint, we can set $p_{p}{\left(Civ.\right)}_{Com.}=1\times 10^{-6}\%$ and finally the proportion of stellar systems hosting extant intelligent civilisations able to communicate by radio signal at the same time is:(27)$$P_{Life}{\left(Civ.\right)}_{Com}=0.1\times 0.22\times 0.75\times 0.94\times 0.19\times 0.00000001=2.9\times 10^{-9}\%$$

Following Equations (Equation 2) and (Equation 27), the associated number of civilisations in the Milky Way is thus:(28)$$N_{Life}{\left(Civ.\right)}_{Com}=6$$

The average distance between two civilisations able to communicate in the Milky Way is then given by Equation (Equation 4) and (Equation 28):(29)$$D_{Life}{\left(Civ.\right)}_{Com}=19,979\;\mathrm{ly}$$

This distance is very large. With signal travelling at the speed of light, it would take 39,958 years for a civilisation to receive an answer from another civilisation, a time frame much greater than the 100 years of duration of the radio communicating civilisation considered in the model. Thus, the probability to detect a radio-signal is very low and if one is detected one day it is possible that the emitting civilisation has since disappeared. This result could explain the Fermi paradox: even if the probability of the existence of extraterrestrial civilisations is relatively high, due to the distance between them, it is more or less impossible for them to communicate with one another. Using our model, an intelligent civilisation must be able to communicate in space for at least about 8937 years to have a chance to receive an answer from another civilisation; $D_{Life}{\left(Civ.\right)}_{Com}$ would then be approximately equal to 4468.5 ly.

### 5.4. Search for Past Extraterrestrial Civilisations

Even if traces of past civilisations could be relatively common in the Milky Way, as indicated by Equation (Equation 18), unless they reached a high stage of evolution associated with large signatures such as those described in [71], the only way to detect such relics would be in situ exploration, which is totally incompatible with the distance of 54 ly given by Equation (Equation 19). We have thus to consider the biosignatures associated with self-destructive civilisations envisaged by [71] (e.g., nuclear detonation, stellar pollution or total planetary destruction). Indeed, some of these biosignatures may remain and be detectable by spectroscopic techniques for more than 100,000 years.

In this case we can set $p_{p}{\left(Civ.\right)}_{S-D}=1\times 10^{-3}\%$ and finally the proportion of stellar systems presently hosting detectable remains of self-destructive civilisations is:(30)$$P_{Life}{\left(Civ.\right)}_{S-D}=0.1\times 0.22\times 0.75\times 0.94\times 0.19\times 0.00001=2.9\times 10^{-6}\%$$

Following Equations (Equation 2) and (Equation 30), the associated number of self-destructive civilisations in the Milky Way is thus:(31)$$N_{Life}{\left(Civ.\right)}_{S-D}=5894$$

The average distance between signature of self-destructive civilisations in the Milky Way is then given by Equations (Equation 4) and (Equation 28):(32)$$D_{Life}{\left(Civ.\right)}_{S-D}=1998\;\mathrm{ly}$$

This distance is relatively large but some of the biosignatures described in [71] could be detected.

### 5.5. Summary

The results of Equations (Equation 20)–(Equation 32) are reported in Table 2. Once again, these values must be treated with caution but they provide an indication of the type of extraterrestrial life that could be detected, its frequency, and the limitations engendered by the available detection techniques.

## 6. From Complex Modeling to General Public Outreach

### 6.1. Describing Important Astronomical Notions

When engaging with general public, it is first necessary to illustrate astronomical distances. An effective way of so doing is to use a model at scale. For instance, using a tennis ball as the Sun, it is possible to represent both the size of the planets and their distances from the Sun as displayed in Table 3.

It is shown that, at this scale, Earth corresponds to a dot of half a millimetre located 7 meters from the Sun. More interestingly, it is shown that at this scale the closest star from the Sun (Proxima Centauri) corresponds to a 9 mm sphere located 1977 km far from the Sun, i.e., the distance from Algiers to Dublin or from Chicago to Miami. Thus, even if the distance between each planet of the Solar System appears high, they are very small in comparison with the distance separating stellar systems. This has a strong consequence on the exploration of space using in situ probes. Indeed, the order of magnitude of the speed of present spacecraft is about 20 km·s^−1^. At that speed, taking a straight line, and in the best trajectory configuration, it requires 5 h and 30 min to reach the Moon, 45 days for Mars, 1 year for Jupiter, 6 years and 326 days for Neptune and 67,272 years for Proxima Centauri. This short demonstration clearly illustrates that the in situ exploration of extraterrestrial bodies by humans is presently limited to the Solar System.

The notion of spacetime is also very important. The speed of light in vacuum *c* is constant and equal to approximately 300,000 km·s^−1^. By constant, one means that the simple velocity vector addition formula does not apply: whatever the referential frame, the speed of light remains the same. The only way to explain this phenomenon is to introduce the notion of time dilatation, expressed by Albert Einstein. This time dilatation is associated with a contraction of lengths and an increase of energy in such a way that no body with mass can approach the speed of light. Moreover, any electromagnetic wave, including light as well as radio waves, cannot exceed the speed of light. At minimum, to reach the Earth, a radio message would require 1.3 s from the Moon, 4 min 20 s from Mars, 35 min from Jupiter, 4 h from Neptune and finally 4 years, 88 days, 16 h 40 min from Proxima Centauri.

Since the speed of light is constant, it is useful to define the light-year unit corresponding to the distance travelled by light in vacuum in one year (365.25 days). It is approximately equal to 10,000 billion km. Proxima Centauri is thus 4.243 ly from the Sun.

### 6.2. Introducing the Probabilistic Approach

The initial goal of this paper was to illustrate the challenge of astrobiology to general public. The model outlined previously is too complex to be described in its entirety and is above all very speculative. However, it permits the introduction of notions pertaining to the origin of life and evolution, the major dichotomy existing between microbial and macroscopic multicellular life in terms of habitability requirements, life detection methods and their limits for space exploration due to technical limitation and/or astronomical distances.

At the centre of this reflection is the fact that life is characterised by evolutionary processes, starting from auto-replicating organic molecules and finishing by becoming intelligent life (even artificial intelligence). It is now accepted that the theory of spontaneous generation is totally obsolete. Thus, it is possible to explain that, to reach the stage of intelligent organisms, life must necessarily pass through important stages that are development of cells (compartmentalisation), photosynthesis (metabolism), and multi-cellularity (complexity). Implicitly, this chronology implies that the further advanced an organism is from the origin of life, the less it is probable in the Universe.

### 6.3. From Intelligent to Microbial E.T.

It is obvious that the general public is more interested by extraterrestrial intelligence than by extraterrestrial microbes. When approaching the notion of extraterrestrial life with the public, it is thus better to start with ‘intelligence’.

Firstly, it is important to say that science considers extraterrestrial intelligence as a possibility. One can highlight the Pioneer Plaque and the Voyager Golden Record. One may focus on two ideas: that (i) the more life is evolved, the less it is probable in the Universe, and (ii) that to reach the stage of intelligent life, it is necessary that the surface of the planet upon which it developed was located in the habitable zone of its star for several billion years. Consequently, from these ideas, we can state that, if they exist, intelligent civilisations are certainly very rare in the Universe, and statistically far from each other. The only way to detect them is thus to search for a radio or light signal, as hypothesised by Frank Drake and developed by the SETI institute with the Arecibo radiotelescope. However, using Drake’s equation, or Equation (Equation 27) herein, we can hope that there are presently circa six intelligent civilisations in the Milky Way and, since there are approximately 350 billion galaxies, several hundred billion intelligent civilisations in the Universe. Assuming that these civilisations are homogeneously spread within our galaxy means that they are separated by approx. 20,000 ly (Equation (Equation 29)). Taking into account Einstein’s special relativity as described above, this means that physical contact between these civilisations is virtually impossible. Moreover, this also means that a message send into space will require 20,000 years to reach a planet inhabited by intelligent E.T. life and that this message would receive an answer not less than 40,000 years after it was sent. Over such a timescale, it is possible that the civilisation which sent the initial message would have disappeared before the answer arrives. Exchange with an extraterrestrial civilisation using radio communication is thus also very unlikely.

On the other hand, the number of civilisations in the Milky Way reaches about 12,000 when considering intelligent civilisations not sufficiently developed to communicate in space, and more than several tens of millions when considering past civilisations (Equations (Equation 11) and (Equation 17)). This means that planets inhabited by ‘primitive’ civilisations could exist at distances of only 1,586 ly (Equation Equation 13) and that traces of past civilisations could be present at only 54 ly (Equation (Equation 19)), i.e., over ‘human timescales’. However, the associated biosignatures would require in situ exploration to be detected, and they are too distant to be examined during human lifetimes. Even if less probable, a solution may be to detect past self-destructing civilisations due to the potentially long time duration for which their detectable remnants may endure [71].

It is therefore necessary to consider what we can detect from Earth. We can now study the atmosphere of an exoplanet and attempt to detect the spectral signature of life’s activity at its surface (biologically produced gases, for instance). Using our model, we obtained an average distance between the Earth and a planet inhabited by photosynthetic life of 41 ly, and of 78 ly for a planet colonised by vegetation (Equations (Equation 22) and (Equation 26)). These distances are compatible with present and near-future instruments such as the Kepler or PLATO space telescopes.

Microbial life does not have the same habitability requirement to develop. The notion of the traditional habitable zone is not applicable in this case, since habitable environments where microbial life may exist at depths of several kilometres in the ocean or in the crust. This means that the subsurface of Mars and the icy moons of Jupiter and Saturn could be inhabited. The problem in this case is that exploring these subsurface environments requires the kind of deep drilling very difficult to make in the near future.

Finally, it is also possible to look for past traces of life at the surface of Mars. This is the aim of the future ESA-Roscosmos mission ExoMars 2020 and NASA Mars 2020 mission. Indeed, during the Noachian (≈3.5 Ga ago), the surface of Mars was habitable, with the presence of liquid water demonstrated by numerous previous missions. So life may have appeared and developed in certain particular areas at the surface before disappearing and/or migrating into the deep aquifer. Plate tectonics appears to have been very limited on Mars, and non-existent at present. We can thus expect that, if microbes were present at the surface of Mars several billion years ago, their remains would have been preserved from tectonic metamorphism and that microfossils could still be found at the surface. By analogy, we can expect to find structures relatively similar to those found in Archaean terrestrial rocks: micrometric cells of simple morphology (coccoidal, rod-shaped, filamentous) made of kerogen (insoluble carbonaceous matter). The difficulty is that, even on Earth using laboratory equipment, demonstrating the biotic origin of these structures is very challenging. It requires complementary geological, mineralogical and elemental evidence obtained using a wide range of techniques, including high-resolution and sophisticated instrumentation currently incompatible with the very limited payloads of in situ space probes. The NASA Mars 2020 mission therefore has the explicit objective of caching samples in preparation for a future sample return mission in order to analyse them in terrestrial laboratories around 2035.

Figure 3 shows the distribution of the number of biosignature detection techniques available and the increasing probability of extraterrestrial life versus its distance from the Earth. Note that the probability of extraterrestrial life and the stage of evolution that can be reached versus the distance from the Earth follow a similar evolution. The strong decrease observed in the number of available techniques is due to the impossibility of carrying out in situ exploration outside the Solar System. Figure 3 permits us to illustrate how the concept of detectability constitutes the main challenge for proving the existence of the extraterrestrial life. While it is technically possible to detect microbial life in the Solar System, the probability of existence of this life is very low. On the other hand, if the probability of existence of evolved life out of the Solar system is very high, the detection techniques available to identify these biosignatures are very limited. The limiting criterion for the detection of extraterrestrial life is thus the low probability of the existence of such life at short distances and the restricted number of available detection techniques far from the Earth. Finally, Figure 3 explains the fact that, in our study, photosynthetic life appears to be the form of life that is the most probable to be detected; the average distance from the Earth being statistically compatible with the associated detection techniques (Table 2).

### 6.4. Testing the Approach with Teachers

In the framework of the Maison Pour La Science, in Orléans, France, the Exobiology team of the CBM has organised 2 days of training courses about exobiology for teachers from primary schools and secondary schools (the French ‘collège’). During these two days, all the aspects of the domain of astrobiology were tackled, from prebiotic chemistry, Earth-life evolution, the conditions of the primitive Earth, micropalaeontology, space instrumentation, exploration and search for extraterrestrial life. The aim of the training course is to give to the teachers a general background on the subject and to present them with easy tasks and projects that can be performed at school with their students (aged 6–15). For the courses concerning the search for extraterrestrial life, we used the approach proposed in Section 6.1, Section 6.2 and Section 6.3 of this paper.

Then, in order to illustrate the consequences of the different points we proposed a task in which the students have at their disposal:

(i) an information sheet containing practical information (the speed of light, the definition of a lightyear, the size of the Milky Way, the speed of present spacecraft and the different ingredients necessary for life);

(ii) several information sheets describing different celestial bodies (the Moon, Mars, Enceladus, Gliese 667 Cc, Kepler-438b, and a fictional planet X28-42), types of mission (rover, sample return, satellite, optical telescope, and radiotelescope), and types of life (microfossils, microbes, photosynthetic organisms, complex multicellular organisms, ruins of a past intelligent civilisation, and a present extraterrestrial civilisation able of telecommunication).

Each celestial body is described by its distance from the Earth, its surface temperature and pressure, and its habitability conditions. For the fictional planet X28-42, fictional values and complementary information are used (e.g., it is located at 20,000 ly from the Earth and has an intelligent civilisation at its surface, which has been capable of telecommunication for more than 10,000 years). Each type of mission is described, including the types of biosignatures that can be detected with that mission, and the distance of detection is given. Additionally, for each stage of life, the minimal requirement for its appearance and preservation is given, as well as its size, the associated biosignatures and the probability of its existence in the Milky Way (from very likely to almost zero).

Finally, using this information, the students must complete a chart in which, for each body, they must identify the usable types of mission, the possible traces of life, the required type of mission, and the detectable traces of life (corresponding to the result of the three first parameters). They have then to make a conclusion for each body. In our exercise, it was concluded (i) that microfossils could be detected at the surface of Mars but that it will probably require a sample return mission, (ii) that we could detect photosynthetic life by analysing the atmosphere of Kepler-438b, and (iii) that the detection of traces of life is not possible on any of the other bodies, either for technical and/or astronomical reasons or because life could not appear. For instance, life is not possible on the Moon, and the exploration of Enceladus’ subsurface ocean is not possible at present. Moreover, despite the fact that photosynthetic life is implied on Gliese 667Cc, within its fictitious construct in the task, the planet does not transit in front of its star, and therefore the signal of oxygenic gases will not be remotely detected. Finally, the radio signal coming from X28-42 will arrive on the Earth in no less than 10,000 years, making its detection on human timescales unfeasible.

The reception from the teachers on the approach and on the exercise was very high. The survey filled by the participants at the end of the 2 days gave a satisfaction rate of 95%.

## 7. Conclusions

We have used a more or less empirical statistical approach to evaluate the probability of the existence of extraterrestrial life as a function of its stage of evolution. The results permit determination of the order of magnitude of the distance separating these forms of life from the Earth. Depending on this distance, we discussed the potential of detectable biosignatures. Finally, we have shown that the more extraterrestrial life has evolved, the greater its likely distance from the Earth, and thus the lower the possibility of its detection. Consequently, if extraterrestrial civilisations are possible, we could only detect them at present if they are able to communicate in space using a radio or a light signal. Similarly, even if the probability of microbial life in other stellar systems is relatively high, we could only detect it on habitable planets if it has reached a sufficient stage of evolution to release gases that modify the atmosphere of the planet. Finally, even if we are now able to make in situ investigations on several potentially inhabited bodies of the Solar System, these investigations are still limited to surface exploration. The only current target on which we can search for extra-terrestrial life is thus Mars.

Despite these apparently discouraging conclusions for the possibility to detect extraterrestrial life, this study also demonstrates that life is probably common in the Universe. Moreover, even if the probability to detect extraterrestrial intelligence is close to zero, the probability to detect photosynthetic life on exoplanets is not at all negligible. The probability to find active microbial life on a body other than Earth in the Solar System is also high enough to justify future projects involving investigation of the icy moons of Saturn and Jupiter, as well as the present and future missions to Mars. The approach outlined herein has been used to simply and successfully explain the challenge of astrobiology to the general public, in particular during a two day training course for teachers, and appeared to be very useful and well-understood.

## Figures and Tables

**Figure 1 life-07-00040-f001:**
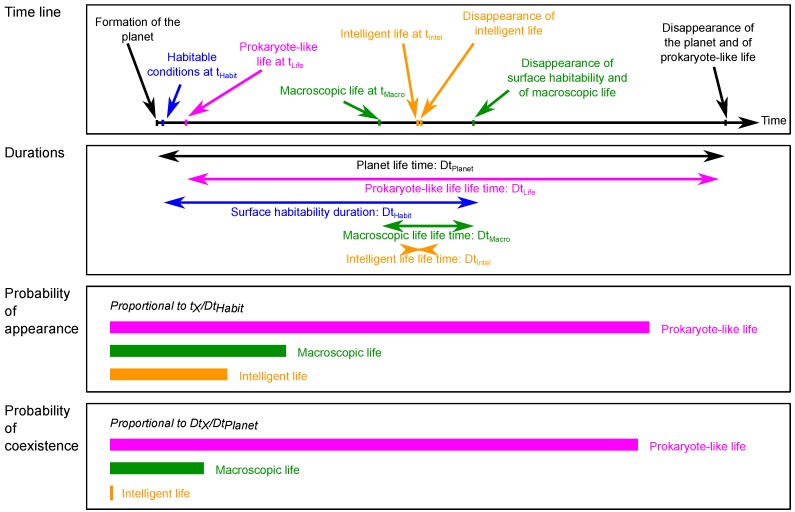
Illustration of the statistical approach used in this study to estimate the probability of appearance and of coexistence of life at different stages of evolutions on a habitable planet based on the Earth life time scale.

**Figure 2 life-07-00040-f002:**
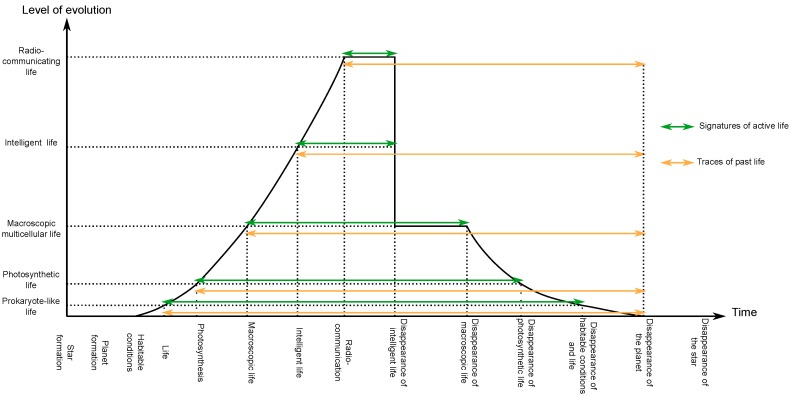
Schematic evolution of life on a habitable planet from the origin to the disappearance of its stellar system. The arrows correspond to the duration of existence of active life (green), and the duration of traces of past life (yellow) associated to different stages of evolution.

**Figure 3 life-07-00040-f003:**
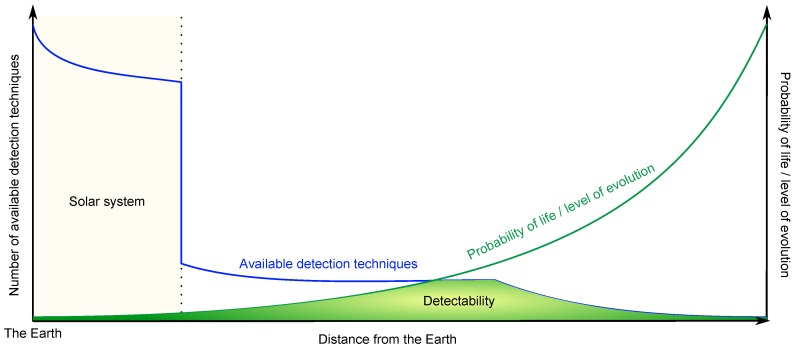
Schematic evolution of the number of biosignature detection techniques available and of the probability of extraterrestrial life versus the distance from the Earth. The stage of evolution follows a similar curve to that of the probability of extraterrestrial life. The detectability area corresponds to the area where the detection techniques are compatible with the associated considered forms of life.

**Table 1 life-07-00040-t001:** Values of the different parameters pi used for the solving of Equation (Equation 1), for the different ‘stages’ of life considered (see main text). $P_{Life}$ is the probability that a stellar system hosts the associated traces of life and $N_{Life}$ is the corresponding number of bodies in the Milky Way. Finally, $D_{Life}$ is the average distance in light years between two of these bodies (i.e., average distance from the Earth).

Type of Life	$p_{S}$	$p_{B}$	$p_{C}$	$p_{L}$	$p_{R}$	$p_{P}$	$p_{\mathrm{Life}}$ (%)	$N_{\mathrm{Life}}$	$D_{\mathrm{Life}}$ (ly)
Prokaryote-like life	1.00	1.00	0.22	0.17	1.00	0.90	3.36%	6.73 × 10^9^	19
Macroscopic multicellular life	0.10	0.22	0.75	0.94	0.28	0.16	0.07%	1.39 × 10^8^	70
Present civilisation	0.10	0.22	0.75	0.94	0.19	0.00002	0.000006%	11,788	1586
Past macroscopic multicellular life	0.10	0.22	0.75	0.94	0.28	0.56	0.24%	4.86 × 10^8^	46
Past civilisation	0.10	0.22	0.75	0.94	0.19	0.50	0.15%	2.95 × 10^8^	54

**Table 2 life-07-00040-t002:** Values of the different parameters $p_{i}$ used for the solving of Equation (Equation 1), for the different kinds of life considered, taking into account their detectability from the Earth or by using in situ investigation (see main text). $P_{Life}$ is the probability that a stellar system or the considered body hosts the considered traces of extra-terrestrial life. Finally, $N_{Life}$ is the corresponding number of bodies in the Milky Way and $D_{Life}$ is the average distance in light years between these bodies (i.e., average distance from the Earth).

Type of Life	$p_{S}$	$p_{B}$	*$p_{C}$*	$p_{L}$	$p_{R}$	$p_{P}$	$p_{\mathrm{Life}}^{2\mathrm{nd}}$	$p_{\mathrm{Life}}$ (%)	$N_{\mathrm{Life}}$	$D_{\mathrm{Life}}$ (ly)
Extant extraterrestrial										
radio-communicating	0.10	0.22	0.75	0.94	0.19	1 × 10^8^	N.A.	2.9 $\times 10^{-9}\%$	6	19,979
life										
Past extraterrestrial										
self-destructing	0.10	0.22	0.75	0.94	0.19	1 × 10^5^	N.A.	2.9 × 10^−6^%	5894	1998
civilisation										
Vegetation	0.10	0.22	0.75	0.94	0.23	0.14	N.A.	0.05%	10 × 10^7^	78
Photosynthetic life	0.10	0.22	0.75	0.94	0.64	0.34	N.A.	0.34%	6.75 × 10^8^	41
Extraterrestrial										
microbial life	1.00	1.00	0.22	0.17	1.00	0.90	0.337	0.11%	N.A.	N.A.
in the Solar System										
Microfossils on Mars	1.00	0.22	0.75	0.94	1.00	0.95	0.337	0.05%	N.A.	N.A.

**Table 3 life-07-00040-t003:** Illustrating the astronomical distances using a tennis ball to represent the Sun.

		If the Sun Had the Size of	True Distance	If the Sun Had the Size of
	Diameter in km	a Tennis Ball, the Diameter	from the Sun	a Tennis Ball, the Distance
		Would Be in mm	in km	Would Be in m
Sun	1,395,200	65	0	0.0
Mercury	4900	0.23	58,000,000	2.7
Venus	12,000	0.56	108,000,000	5.0
The Earth	12,800	0.60	150,000,000	7.0
The Moon	3474	0.16	150,000,000	7.0
Mars	6800	0.32	228,000,000	10.6
Jupiter	140,000	6.52	780,000,000	36.3
Saturn	120,000	5.59	1,430,000,000	66.6
Uranus	52,000	2.42	2,880,000,000	134.2
Neptune	50,000	2.33	4,497,000,000	209.5
Proxima Centauri	200,000	9.32	42,430,000,000,000	1977 km

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
