# Peer review of "A Statistical Approach to Illustrate the Challenge of Astrobiology for Public Outreach"

_life, 2017, doi:10.3390/life7040040_

Round 1

Reviewer 1 Report

This article outlines a highly simplified statistical approach to demonstrate that searching for prokaryotic or other microbial forms of life is more likely to be successful than searching for extraterrestrial intelligence (SETI), using an approach heavily influenced by Drake’s approach in formulating the Drake Equation.

I understand that this study is for illustrative purposes, and its intent is to be used as outreach rather than research, but I would like to see the authors clarify a few points before publication:

GENERAL COMMENTS

i) Throughout, the authors give the impression that SETI is only achievable via radio searches.  This is patently untrue - searches for optical signals have been recommended as early as radio (Cocconi & Morrison’s 1959 paper recommending radio SETI was very quickly followed by Schwartz & Towne (1961)’s recommendation of searching for “optical masers”).

It is also the case that exoplanet searches can assist SETI, either in the detection of “technomarkers” (e.g. pollutant molecules such as CFCs in atmospheres), or in the aberration of light curves due to megastructures (cf KIC 862582 or “Tabby’s Star”).  There is a growing literature outlining various strategies that directly utilise exoplanet data - I would like to see some small fraction of this literature cited

In short, I would like to see this paper reflect the current state of the art in SETI, i.e. a multi wavelength endeavour that, while principally conducted in radio/microwave wavelengths, has been rolled out to a wide range of frequencies.

ii) Assuming the probability of appearance of a type of life is inversely proportional to the time it took to arrive is an acceptable assumption for outreach, but in an academic output the authors must outline the  arguments against this more clearly.  I am thinking specifically of 

http://www.pnas.org/content/109/2/395.full

I would like to see the authors acknowledge and respond to this work.

SPECIFIC COMMENTS

i) Line 112: Ozmo —> Ozma

ii) Line 202: “More or less all stellar systems … are capable of supporting unicellular life”.  Are the authors willing to state that even massive stars, with lifetimes of only a few million years, host life? These are admittedly a small fraction of the total, but I think this statement needs to be carefully qualified.

iii) Line 235: "Of the confirmed exoplanets in the HZ, 65% are compatible with complex life”.  This is a massively controversial statement, and assumes a great deal about the surface properties of these bodies (of which we know next to nothing).  For example, our current datasets do not distinguish between Earth-like and Venus-like exoplanets!

The authors need to be explicit about the *very* large assumptions made to reach this value.

iv) Line 334: Dead civilisations leave very different signatures to detect compared to their living cousins:

http://adsabs.harvard.edu/abs/2016IJAsB..15..333S

The authors should discuss how the signatures of dead civilisations decay both in space and time compared to living civilisations.

v) Line 380: Large amounts of O2 and O3 can be generated abiotically on planets orbiting low mass stars! The authors should note that the problem of false positives in biosignature detection is a large one, in particular for planets around low mass stars, which will be the first to be fully characterised.

Author Response

Dear reviewer

Thank you for your interest and for your useful comments on our manuscript.

Here are the answers to your comments and suggestions:

i) Throughout, the authors give the impression that SETI is only achievable via radio searches.  This is patently untrue - searches for optical signals have been recommended as early as radio (Cocconi & Morrison’s 1959 paper recommending radio SETI was very quickly followed by Schwartz & Towne (1961)’s recommendation of searching for “optical masers”).

It is also the case that exoplanet searches can assist SETI, either in the detection of “technomarkers” (e.g. pollutant molecules such as CFCs in atmospheres), or in the aberration of light curves due to megastructures (cf KIC 862582 or “Tabby’s Star”).  There is a growing literature outlining various strategies that directly utilise exoplanet data - I would like to see some small fraction of this literature cited

In short, I would like to see this paper reflect the current state of the art in SETI, i.e. a multi wavelength endeavour that, while principally conducted in radio/microwave wavelengths, has been rolled out to a wide range of frequencies.

Section 5.3 dedicated to the SETI has been improved and the recommended references have been added. The term ‘communicate by radio signal’ has been changed to ‘communicate in space’ throughout the manuscript.

ii) Assuming the probability of appearance of a type of life is inversely proportional to the time it took to arrive is an acceptable assumption for outreach, but in an academic output the authors must outline the  arguments against this more clearly.  I am thinking specifically of 

http://www.pnas.org/content/109/2/395.full

I would like to see the authors acknowledge and respond to this work.

We were not aware of this very interesting paper. This study is based on a similar approach as ours, i.e. linking the probability of an event to occur to the time it spent to occur. We have added it to our references. More generally, we also added a discussion about the peculiarity of the evolution of life on Earth in the beginning of the part 4.2.

i) Line 112: Ozmo —> Ozma

Done.

ii) Line 202: “More or less all stellar systems … are capable of supporting unicellular life”.  Are the authors willing to state that even massive stars, with lifetimes of only a few million years, host life? These are admittedly a small fraction of the total, but I think this statement needs to be carefully qualified.

Even if the presence of extraterrestrial life is not probable around massive blue giants due to their very short lifetime, exoplanets have been found around them and these systems can thus potentially host habitable niches without meaning that life had the time to appear (which is not the concern of this paragraph). A few lines have been added to this effect in the manuscript.

iii) Line 235: "Of the confirmed exoplanets in the HZ, 65% are compatible with complex life”.  This is a massively controversial statement, and assumes a great deal about the surface properties of these bodies (of which we know next to nothing).  For example, our current datasets do not distinguish between Earth-like and Venus-like exoplanets!

The authors need to be explicit about the *very* large assumptions made to reach this value.

This is totally true. Indeed, we choose the so called ’mesoplanets’ as the only planets where complex life may appear. We developed this part and update this value according to https://fr.wikipedia.org/wiki/Liste_d’exoplanètes_potentiellement_habitables. The proportion is now 75% instead of 65%. We added a comment to say that it is very speculative.

iv) Line 334: Dead civilisations leave very different signatures to detect compared to their living cousins:

http://adsabs.harvard.edu/abs/2016IJAsB..15..333S

The authors should discuss how the signatures of dead civilisations decay both in space and time compared to living civilisations.

It is totally true and we forget to mention this possibility. We have added section 5.4 dedicated to this hypothesis.

v) Line 380: Large amounts of O2 and O3 can be generated abiotically on planets orbiting low mass stars! The authors should note that the problem of false positives in biosignature detection is a large one, in particular for planets around low mass stars, which will be the first to be fully characterised.

This is also true and has been added in the manuscript with the appropriate reference.

Thank you again

Best regards

Frédéric Foucher

Reviewer 2 Report

p { margin-bottom: 0.25cm; line-height: 120%; }

In this study, the authors develop a simple statistical tool, similar to the Drake equation, to provide order of magnitude estimates for the commonness of life at different evolutionary stages in the universe. From this they estimate the distance from Earth to the nearest example of a planet / moon hosting microbial, macroscopic, or intelligent life (either extant or extinct) and weigh this against the prospects of detecting it. This work builds upon previous statistical analyses of the occurrence of life / habitable planets / detectable biosignatures to frame the challenges in the field of astrobiology in a novel and accessible way for a public audience.

There are many assumptions and simplifications made in this approach, which the authors acknowledge in the text. However, this is a reasonable approach given that the aim here is to produce a public outreach tool, which, like the Drake equation, frames the problem rather than providing a detailed solution. I believe this a very useful outreach and teaching tool that will help astrobiologists communicate the challenges and future expectations of their work. I would recommend this for publication once a few minor comments / clarifications have been addressed.

Minor Comments:

Equation 3:

Missing volume units.

Line 209:

The use of the term “yellow dwarf” is misleading, because this tends to refer only to G-type stars similar to the Sun. The values you use (10% for the proportion of stars in the galaxy and the 22% value from Petigura et al.) refer to a broader category of “sun-like stars”, which refers to G- and K-type stars (and sometimes F-type stars). I would recommend replacing “yellow dwarf stars” with “sun-like stars” in the text.

Line 235:

How is the 65% value determined? Assuming “compatible with complex life” means planets orbiting G- and K-type stars, from the tables of habitable zone (HZ) exoplanets I get around 40% when looking at conservative+optimistic confirmed HZ planets. Also, it would be useful to include the access date for the website from which the information was sourced; the habitable exoplanet list will update regularly.

Line 274-5:

Life appearing on one of six potential bodies in the solar system should result in a probability of ~16%, not 14%.

Line 430:

Note that surface features like the vegetation red edge would be even more challenging to detect than atmospheric features.

Lines 440-441:

I agree that the distance is reasonable for detection; however, this will be a job for upcoming and planned telescopes, rather than Hubble. Hubble is only really able to give us information about the atmospheres of some giant planets. Larger telescopes than Hubble would be needed to characterize HZ Earth-like planets. For atmospheric characterization of HZ rocky planets, telescopes like JWST could detect some biosignature gases (depending on distance, gas concentration, atmospheric density, etc.), assuming the planet transits its host star. Direct imaging of a planet is needed to detect surface signatures like the vegetation red edge. Limited direct imaging of the surfaces of nearby, rocky planets may be possible with JWST; however, more powerful telescopes would be needed to really open the door to such observations.

Author Response

Dear reviewer

Thank you for your interest and for your useful comments on our manuscript.

Here are the answers to your comments and suggestions:

Equation 3:

Missing volume units.

Done

Line 209:

The use of the term “yellow dwarf” is misleading, because this tends to refer only to G-type stars similar to the Sun. The values you use (10% for the proportion of stars in the galaxy and the 22% value from Petigura et al.) refer to a broader category of “sun-like stars”, which refers to G- and K-type stars (and sometimes F-type stars). I would recommend replacing “yellow dwarf stars” with “sun-like stars” in the text.

Done

Line 235:

How is the 65% value determined? Assuming “compatible with complex life” means planets orbiting G- and K-type stars, from the tables of habitable zone (HZ) exoplanets I get around 40% when looking at conservative+optimistic confirmed HZ planets. Also, it would be useful to include the access date for the website from which the information was sourced; the habitable exoplanet list will update regularly.

This is totally true. Indeed, we choose the so called ’mesoplanets’ as the only planets where complex life may appear. We developed this part and update this value according to https://fr.wikipedia.org/wiki/Liste_d’exoplanètes_potentiellement_habitables. The proportion is now 75% instead of 65%. We added a comment to say that it is very speculative.

Line 274-5:

Life appearing on one of six potential bodies in the solar system should result in a probability of ~16%, not 14%.

This value has been changed as well as the results of all further equation where this value was used.

Line 430:

Note that surface features like the vegetation red edge would be even more challenging to detect than atmospheric features.

This comment has been added in the manuscript.

Lines 440-441:

I agree that the distance is reasonable for detection; however, this will be a job for upcoming and planned telescopes, rather than Hubble. Hubble is only really able to give us information about the atmospheres of some giant planets. Larger telescopes than Hubble would be needed to characterize HZ Earth-like planets. For atmospheric characterization of HZ rocky planets, telescopes like JWST could detect some biosignature gases (depending on distance, gas concentration, atmospheric density, etc.), assuming the planet transits its host star. Direct imaging of a planet is needed to detect surface signatures like the vegetation red edge. Limited direct imaging of the surfaces of nearby, rocky planets may be possible with JWST; however, more powerful telescopes would be needed to really open the door to such observations.

The conclusion of section 5.2 has been changed to take into account this comment.

Thank you again

Best regards

Frédéric Foucher